# The Pharmacokinetics of CPZEN-45, a Novel Anti-Tuberculosis Drug, in Guinea Pigs

**DOI:** 10.3390/pharmaceutics15122758

**Published:** 2023-12-12

**Authors:** Lucila Garcia-Contreras, Shumaila Nida Muhammad Hanif, Mariam Ibrahim, Phillip Durham, Anthony J. Hickey

**Affiliations:** 1Department of Pharmaceutical Sciences, College of Pharmacy, The University of Oklahoma Health Sciences Center, Oklahoma City, OK 73117, USA; 2Department of Biomedical Sciences, Kentucky College of Osteopathic Medicine, University of Pikeville, Pikeville, KY 41501, USA; shumailahanif@upike.edu; 3AstraZeneca, One MedImmune Way, Gaithersburg, MD 20878, USA; mariam.ibrahim@astrazeneca.com; 4Department of Biomedical Engineering, University of North Carolina at Chapel Hill, Chapel Hill, NC 27599, USA; pgdurham@email.unc.edu; 5RTI International, 3040 Cornwallis Road, Research Triangle Park, NC 27709, USA; ahickey@unc.edu

**Keywords:** tuberculosis, insufflation, passive inhalation, pharmacokinetics studies, CPZEN-45, guinea pigs

## Abstract

CPZEN-45 is a novel compound with activity against drug-susceptible and drug-resistant tuberculosis (TB). The present study was undertaken to determine the best dose and dosing regimen of inhalable CPZEN-45 powders to use in efficacy studies with TB-infected guinea pigs. The disposition of CPZEN-45 after intravenous, subcutaneous (SC), and direct pulmonary administration (INS) was first determined to obtain their basal pharmacokinetic (PK) parameters. Then, the disposition of CPZEN-45 powders after passive inhalation using consecutive and sequential doses was evaluated. Plasma concentration versus time curves and PK parameters indicated that the absorption of CPZEN-45 after INS was faster than after SC administration (Ka = 12.94 ± 5.66 h^−1^ and 1.23 ± 0.55 h^−1^, respectively), had a longer half-life (2.06 ± 1.01 h versus 0.76 ± 0.22 h) and had higher bioavailability (67.78% and 47.73%, respectively). The plasma concentration versus time profiles and the lung tissue concentration at the end of the study period were not proportional to the dose size after one, two, and three consecutive passive inhalation doses. Three sequential passive inhalation doses maintained therapeutic concentration levels in plasma and lung tissue for a longer time than three consecutive doses (10 h vs. 3 h, respectively). Future studies to evaluate the efficacy of inhaled CPZEN-45 powders should employ sequential doses of the powder, with one nominal dose administered to animals three times per day.

## 1. Introduction

Tuberculosis (TB) is the second cause of death from an infectious disease after COVID-19. The number of deaths caused by TB increased from 1.2 million in 2019 to 1.3 million in 2020, and this number is likely higher due to limited drug susceptibility testing of patient TB samples due to the COVID-19 pandemic [1]. In fact, the increase of TB cases in Washington state in the United States in early 2022 has been called “The worst TB outbreak in 20 years” [2,3]. There have been two important developments in the treatment of drug-susceptible and drug-resistant TB (multidrug-resistant (MDR) and extensively drug-resistant (XDR) TB) in the last decade. The first is the emergence of the Nix regimen of pretomanid, bedaquiline, and linezolid [4]. This is the first entirely new regimen for TB therapy to appear in over a half-century and is cause for optimism in controlling the global burden of the disease. However, it is understood that this regimen has drawbacks, not least of which is the toxicity of linezolid [5]. In addition, the World Health Organization recently announced that injectable medicines should be phased out in all regimens for the treatment of drug-resistant TB [6]. Despite these two new developments, TB treatment times are still long, and side effects remain the main obstacle to patient compliance. Consequently, there remains an urgent need for the development of effective complementary strategies to existing drug treatments to control the TB global pandemic and shorten treatment courses.

Caprazamycins are a group of novel antimycobacterial agents isolated from a culture broth of *Actinobacteria Streptomyces* [7]. CPZEN-45 (Figure 1) is one of the derivatives that was initially produced by introducing various amines, anilines, and alcohols at the carboxyl group of the diazepinone ring [7]. Like pretomanid, CPZEN-45 inhibits the cell wall biosynthesis of the bacteria, but while pretomanid blocks the oxidation of hydroxymycolate to ketomycolate [8], CPZEN-45 inhibits the bacterial translocase I, which is one of the key enzymes for peptidoglycan biosynthesis [9].

In vitro, CPZEN-45 has a minimum inhibitory concentration (MIC) of 1.56 and 6.5 μg/mL for drug-susceptible MTB and MDR-TB, respectively [10], and showed no toxicity in cell assays [11]. In vivo, it showed therapeutic efficacy in mice infected with extensively resistant TB (XDR-TB) when injected subcutaneously (SC) [7]. However, CPZEN-45 has a low oral bioavailability due to its poor absorption from the gastrointestinal tract [12]. Administration of CPZEN-45 by the pulmonary route as an inhaled aerosolized powder would deliver the drug directly to the primary site of infection, achieving higher drug concentrations in the lungs and potentially reducing treatment time. Thus, direct pulmonary administration is favored over other routes of administration.

Researchers working to bring novel compounds from the bench to the bedside are faced with a conundrum; they either perform comprehensive pharmacokinetic (PK) studies to determine the most effective drug dosing regimen to be used in efficacy studies first or they demonstrate that said drug is effective against the disease. Furthermore, compounds to be administered by the pulmonary route require an optimized formulation to ensure that sufficient drugs will be delivered to the site of action. The inhalable dry powder formulation developed for CPZEN-45 was prepared with the drug alone and has been demonstrated to be dispersible as aerosols with large respirable fractions [13,14]. Thus, a smaller dose of the CPZEN-45 powder for inhalation may be required, compared to other commercially available inhalable antibiotics [15], because this formulation was prepared without excipients [16].

Efficacy studies were then performed in the guinea pig model of TB using the optimized CPZEN-45 formulation and a preliminary dose selected from a brief PK study [17]. Pulmonary administration of CPZEN-45 powder for inhalation resulted in a 1-log reduction in the bacterial burden in the lungs of animals treated with dry powder aerosols of CPZEN-45 or an SC injection of solution compared to untreated animals [18]. One of the factors that may have contributed to this modest result is that the pulmonary dose of CPZEN-45 powders selected from the preliminary PK study was insufficient to decrease further the bacterial burden in affected organs. To address this hypothesis, the present study was undertaken to evaluate different doses of inhalable CPZEN-45 powders and different dosing regimens, with the goal of determining the dose that can maximize the effectiveness of CPZEN-45 to treat TB.

## 2. Materials and Methods

### 2.1. Materials

CPZEN-45 hydrochloride salt (Lot #: NC7-E07149-099) was obtained from the Infectious Disease Research Institute (IDRI, Seattle, WA, USA) and was used as received. Acetonitrile (HPLC grade ≥ 99.5%) and trifluoroacetic acid (TFA) were obtained from Sigma (St. Louis, MO, USA). Ultrapure water was obtained from Pure Lab Ultrapure Water System (ELGA, High Wycombe, UK). CPZEN-45 dry powder for inhalation was manufactured by spray drying [13] at the Research Triangle Institute (RTI International, RTP, Research Triangle Park, NC, USA) and was used as received. CPZEN-45 powder for inhalation consisted of particles with a mean mass aerodynamic diameter (MMAD) of approximately 3 µm and 45% mass below 2.82 µm. Briefly, a 10 mg/mL CPZEN-45 hydrochloride solution was sprayed at a liquid flow rate of 5 mL/min into the drying chamber, having a nitrogen flow rate of 1052 L/h and inlet temperature of 190 °C. Microparticles were collected using a cyclone with an aspiration rate of 35 m^3^/h [13].

### 2.2. Animals

Male guinea pigs (Hartley Dunkin, Charles River, Wilmington, MA, USA) weighing 433.09 ± 36.50 g were used in these studies. The animals were housed in a 12 h light/12 h dark cycle at a constant temperature environment of 22 °C. A standard diet and water were supplied ad libitum. All animal procedures were approved by the Institutional Animal Care and Use Committee (IACUC) of the University of Oklahoma Health Sciences Center (OUHSC). Twenty-four hours before the study, each animal underwent cannulation of the right external jugular vein for continuous blood sampling, as described previously [19].

### 2.3. Treatments

Table 1 summarizes the treatment groups studied and the purpose of each group in the study. These consisted of animals receiving a single dose of CPZEN-45 administered by intravenous (IV), subcutaneous (SC), and insufflation (INS) (groups 1–3) and animals dosed by passive inhalation receiving consecutive (groups 4–6) or sequential doses (group 7) of aerosolized CPZEN-45 powder.

**Table 1 pharmaceutics-15-02758-t001:** Study design for the disposition of CPZEN-45 in guinea pigs after administration of CPZEN-45 solution or powder.

Group #	Formulation—Route of Administration	Dose	Purpose in the Study
1	Solution—IV	1 mg/kg	Obtain basic PK parameters in absence of drug absorption
2	Solution—SC	1 mg/kg	Determine bioavailability of drug after administration by the conventional route
3	Powder—INS	1 mg/kg	Determine pulmonary bioavailability by direct administration of the whole dose to the airways
4	Powder—Passive inhalation	* Single (80 mg) = 8 dosators over 24 min	Determine the size of an inhaled dose (number of dosators) that would maintain therapeutic concentrations for the longest period of time
5	Powder—Passive inhalation	* Double (160 mg) = 16 dosators over 48 min	
6	Powder—Passive inhalation	* Triple (240 mg) = 24 dosators over 72 min	
7	Powder—Passive inhalation	* Three sequential (80 mg) = 8 dosators every 5 h	Determine if there is powder accumulation after a single inhaled dose = 8 dosators

* The dose and dosing regimen for study groups 4–7 is illustrated in Figure 2, as shown below.

#### 2.3.1. Direct Administration of a Single Dose of CPZEN-45 by Different Routes

Animals were randomly divided into three groups; each group received CPZEN-45 at a dose of 1 mg/kg in solution by the IV (*n* = 7) or SC (*n* = 6) routes or CPZEN-45 powders (1 mg/kg) by the pulmonary route (INS; *n* = 6) (Table 1). CPZEN-45 solutions were prepared by dissolving a weighed amount of CPZEN-45 hydrochloride salt in water for injection and filtering the solution through a 0.22-μm Millex GS filter (Millipore, Cork, Ireland). To administer the CPZEN-45 powders by INS, each animal was first anesthetized, and the trachea was visualized with the help of a laryngoscope. The tube of a small-animal insufflator (Penn Century Inc., Wyndmoor, PA, USA) was inserted into the trachea, and the CPZEN-45 powder was aerosolized into the airways of the animal with the help of air from an empty 10 mL syringe [19]. Blood samples (0.3 mL) were collected from each animal into heparinized micro-centrifuge tubes at 0 (prior to dosing), 0.08, 0.25, 0.50, 1, 1.5, 2, 3, 4, and 5 h after administration of the single dose CPZEN-45 by the different routes. Warm sterile saline solution was used to replace the volume of blood drawn after each sample collection to maintain the volume of distribution constant in each animal.

#### 2.3.2. Passive Inhalation of Consecutive or Sequential Doses of CPZEN-45 Dry Powder Aerosols

A custom-made nose-only dosing chamber was designed to dose the animals by passive inhalation (Figure 3a,b) [17]. The design of this chamber, including the inhalation area size, the method of aerosol generation from CPZEN-45 powders, and the manner in which powders were introduced into the inhalation area were first optimized [17,20]. The CPZEN-45 dry powder aerosol was generated by passing air from a 10 mL syringe through a “dosator” that was developed to meter the powder dose. Each dosator was prepared to contain 10 mg of CPZEN-45 powder, with sufficient room in the powder reservoir to fluidize the powder. The tip of the dosator was designed to be introduced into the inhalation area of the chamber (Figure 3c) [20].

The loading dose in the dosing chamber and the length of aerosol exposure were also optimized using a small group of animals with the purpose of matching the area under the CPZEN-45 plasma concentration curve (AUC) obtained after passive inhalation to that after INS [17]. This preliminary study determined that 8 dosators (Figure 3c), aerosolized into the chamber every 3 min, were necessary to match the AUCs by passive inhalation to those after INS [17]. Thus, this regimen was considered as a “single dose by passive inhalation” (Figure 3c).

The next step was to find a dosing regimen, using this “single dose by passive inhalation” that would maintain the CPZEN-45 plasma concentrations above MIC for the maximum length of time while minimizing the time that the animals were held in the ports of the custom nose-only dosing chamber. Four dosing regimens were evaluated: single, double, and triple consecutive doses that could be administered once a day for approximately one hour or three sequential doses administered every 5 h throughout one day (Table 1, Figure 2). The single dosing regimen consisted of 8 dosators that were administered over a period of 24 min (one dosator every 3 min, as described above). In the double dosing regimen, 16 dosators were administered over a period of 48 min (one dosator every 3 min), and in the triple dosing regimen, 24 dosators were administered over a period of 72 min (one dosator every 3 min). The powder in each dosator was aerosolized into the chamber with the help of 10 mL of air from an empty syringe 3–5 times until all powder in the dosator was completely aerosolized. Blood samples (0.3 mL) were collected from each animal into heparinized micro-centrifuge tubes at 0 (prior to dosing), 0.08, 0.25, 0.50, 1, 1.5, 2, 3, 4, and 5 h. To maintain the volume of distribution constant in each animal, warm sterile saline solution was used to replace the volume of blood drawn after each sample collection.

The last group of animals (group 7, *n* = 4) received three sequential “single dose by passive inhalation” doses, with one dose administered every 5 h (Table 1, Figure 2). Under this regimen, the first dose (8 dosators administered over a period of 24 min) was administered at time = 0 h, and animals were left to rest until time = 5 h. The second dose (8 dosators administered over a period of 24 min) was administered at time = 5 h, and animals were left to rest until time = 10 h. The third dose (8 dosators administered over a period of 24 min) was administered at time = 10 h, and animals were left to rest until time = 15 h when they were euthanized. Blood samples (0.3 mL) were collected from each animal into heparinized micro-centrifuge tubes at 0, 0.08, 0.5, 1, 2, 5, 5.083, 5.5, 6, 7, 10, 10.085, 10.5, 11, 12, and 15 h. The volume of blood drawn after each sample collection was replaced with warm sterile saline solution to maintain the volume of distribution constant in each animal.

### 2.4. Sample Collection and Determination of CPZEN-45 in Biological Samples

For all treatment groups, plasma was separated after centrifugation of the tubes containing the blood collected at each time point. After collection of blood at the last time point, animals in all treatment groups were anesthetized and euthanized by exsanguination. To determine the concentration of CPZEN-45 remaining in the airways of each animal at the end of the study period (5 and 15 h, depending on the treatment group; see Figure 2), bronchoalveolar lavage (BAL) was performed as described previously [19]. Afterward, lung and spleen tissues were resected to determine the CPZEN-45 concentrations in these organs that are relevant to TB infection.

Local CPZEN-45 concentrations were also measured in BAL fluid (BALF), spleen, and lung tissue at intermediate time points (0.5 h and 2 h) in animals dosed by INS to compare with plasma concentrations at the same time points. Since BAL is a terminal procedure for small laboratory animals, extra guinea pigs were employed for these time points. All samples were frozen immediately at −80 °C until analysis.

After extraction from plasma, BALF, and tissues, CPZEN-45 concentrations were determined by a validated HPLC method [12]. In plasma samples, the limit of detection (LOD) and limit of quantification (LOQ) for this method were 0.05 μg/mL and 0.29 µg/mL, respectively, whereas in BAL and lung/spleen homogenates, the LOD and LOQ were 0.004 and 0.02 μg/mL, respectively [12].

### 2.5. Data Analysis

#### 2.5.1. Pharmacokinetic (PK) Analysis

CPZEN-45 plasma concentration versus time data were analyzed by non-compartmental methods (Phoenix^®^ WinNonlin v.6; Pharsight Corporation, Mountain View, CA, USA) to obtain the area under the curve (AUC), the apparent total body clearance (CL_ss_), the mean residence time (MRT), the half-life (*t*_1/2_), and the elimination rate constant (K). The maximum CPZEN-45 plasma concentration (*C*_max_) and time to obtain the maximum CPZEN-45 concentration (T_max_) were determined directly from the plasma versus time profiles for each individual animal [19]. The mean absorption time (MAT) was calculated using the differences of MRT for each route of administration as follows:MAT_SC_ = MRT_SC_ − MRT_IV_
(1)
or
MAT_INS_ = MRT_INS_ − MRT_IV_
(2)

Bioavailability after INS (F_pulmonary_) was calculated using the area under the curve from 0 h to *t* (AUC_0-*t*_) values obtained after INS administration using the IV administration as reference (Equation (1)):(3)$$F_{Pulmonary}=\frac{AUC_{INS}}{AUC_{IV}}\times \frac{Dose_{IV}}{Dose_{INS}}$$

Plasma concentration versus time data were further analyzed by fitting the data to a one-compartment body model, first-order absorption, first-order elimination, and no lag time for all treatments to obtain the constant of absorption (K_a_) of CPZEN-45 after SC or INS and to compare the values of K_e_ obtained by non-compartment model analysis. The value of K_a_ was also calculated by the “Residuals” method to rule out the influence of model fitting on the value of K_a_.

PK analysis of CPZEN-45 plasma concentration versus time data from animals in the passive inhalation groups was performed using the absorbed dose instead of the nominal dose (amount of powder aerosolized into the dosing chamber). The absorbed dose after passive inhalation was determined using the pulmonary bioavailability (F_pulmonary_), the IV data, and the AUC after passive inhalation in each group (AUC_aerosol_):(4)$${Dose}_{aerosol}=\frac{\left({Dose}_{IV})({AUC}_{aerosol}\right)}{\left(F_{pulmonary})({AUC}_{IV}\right)}$$

#### 2.5.2. Statistical Analysis

Data from plasma concentration versus time profiles and PK parameters were subjected to analysis of variance (ANOVA, SAS/STAT, Cary, NC, USA), and the difference between experimental groups was determined by the least square difference test. Data for drug concentrations in BALF and lung tissue were analyzed using Graph Pad Prism 8.0 software. A comparison of each treatment was performed using a non-parametric ANOVA (Kruskal–Wallis test) with Dunnett’s multiple comparison test to determine which experimental groups were significantly different from each other. The *p*-values for the primary analysis and the adjusted *p*-values for the multiple comparisons <0.05 were considered statistically significant [21].

## 3. Results

### 3.1. Disposition of a Single Dose of CPZEN-45 after Administration by the IV, SC, and Pulmonary (INS) Routes

The plasma concentration versus time curves after administration of 1 mg/kg CPZEN-45 by the different routes (IV, SC, and pulmonary) are shown in Figure 4. CPZEN-45 was absorbed faster when administered by the pulmonary route, compared to SC injection, as evidenced by the significantly higher (p < 0.0001) plasma concentrations at initial time points (Figure 4).

At the end of the study period, the concentration of CPZEN-45 in plasma also remained higher after INS (0.29 ± 0.18 µg/mL) than when administered by the SC route (0.14 ± 0.14 µg/mL). All routes of administration resulted in parallel and linear terminal phases, indicating that the route and rate of CPZEN-45 elimination are the same regardless of the route of administration.

Figure 5 shows the CPZEN-45 concentration in BALF and lung tissue at 0.5, 2, and 5 h after pulmonary administration (INS). At 0.5 h, the time closer to T_max_,_pulmonary_, the CPZEN-45 concentration in BALF was 0.97 µg/mL and decreased gradually with time, but it was still detectable at the end of the study period (0.14 µg/mL, Figure 5A).

The CPZEN-45 concentrations in the lung tissue (Figure 5B) appeared to mirror those in BALF, suggesting that the dissolution of the drug in the lung epithelial lining fluid and subsequent absorption into lung tissue is gradual. Notably, the CPZEN-45 concentration in lung tissue was almost two-fold higher (3.76 µg/g, Figure 3b) than in plasma at approximately T_max,pulmonary_ (C_max_ = 2.3 µg/mL, Figure 4). Even 2 h after drug administration, drug concentrations remained higher in lung tissue than in plasma (2.7 µg/g of tissue, Figure 5B, versus 0.86 µg/mL, Figure 4).

Figure 6 compares the drug concentrations in BALF and lung tissue at the end of the study for the three different routes of administration.

While very little or no drug could be detected in the BALF of animals dosed by the IV or SC routes, the concentration of drug in the BALF of animals dosed by INS was one-third (0.51 ± 0.11 µg/mL) of the MIC (1.56 µg/mL, Figure 6A). Likewise, no drug was detected in the lung tissue of animals dosed by IV or SC injection at the end of the study period. In contrast, the CPZEN-45 concentration in lung tissue after INS (0.6 ± 0.38 µg/g) was 12-fold higher (p = 0.0001) than the LOD (0.05 µg/mL, Figure 6B). Moreover, even when the drug was not detected in plasma after INS, the CPZEN-45 concentration in lung tissue was one-third (0.6 µg/mL) of the MIC (1.56 µg/mL, Figure 6A). CPZEN-45 was not detected in the spleens of animals after parenteral or pulmonary administration at any time point studied (0.5, 2.0, and 5 h).

The disposition of CPZEN-45 after administration by the different routes is characterized numerically by their PK parameters presented in Table 2. CPZEN-45 was absorbed 2–8 times faster (non-compartmental and compartmental analysis, respectively) when administered by INS compared to SC. Conversely, CPZEN-45 was eliminated 3 times faster after SC injection (K = 0.96 h^−1^) than after INS (K = 0.39 h^−1^; *p* = 0.004), resulting in a significantly longer half-life (t_1/2 =_ 2.06 h; *p* = 0.02) and mean residence time (MRT = 2.83 h; *p* = 0.016) of the drug after pulmonary administration. The area under the curve after INS of CPZEN-45 (AUC_0–∞_ = 5.76 µgh/mL) was slightly larger than after SC administration (4.05 µgh/mL). Consequently, the bioavailability of CPZEN-45 was almost 20% higher for the pulmonary route (F__AUC0-∞_ = 67.78) compared to the suggested SC route (F__AUC0-∞_ = 47.73).

### 3.2. Disposition of CPZEN-45 after Administration of Consecutive Doses by Passive Inhalation (Aerosol)

The plasma concentrations versus time curves after administration of consecutive doses of CPZEN-45 by passive inhalation (aerosol) are compared to those after administration of a single dose by INS in Figure 7. CPZEN-45 appeared to be absorbed faster when administered as a “single dose by passive inhalation” than by INS, as evidenced by the significantly higher (*p* < 0.05) plasma concentrations at initial time points (0.25, 0.5, 1, 1.5, 2, and 3 h) (Figure 7). In contrast, the concentration of CPZEN-45 in plasma at the end of the study appeared to be slightly higher when a single dose was administered by INS compared to passive inhalation. All passive inhalation dosing regimens resulted in parallel and linear terminal phases.

Comparison of the plasma concentrations versus time profiles after single, double, and triple doses of CPZEN-45 powders administered by passive inhalation revealed that CPZEN-45 was absorbed faster and at the same rate at initial time points (0.083, 0.25, 0.5 h) for all three doses. The C_max_ increased proportionally to the dose for the single (6.63 ± 1.80 µg/mL) and double doses (10.30 ± 4.90 µg/mL) but not for the triple dose (7.27 ± 2.32 µg/mL). This lack of proportional increase was also observed at the end of the study period (5 h), where the CPZEN-45 plasma concentration after a double dose (0.65 ± 0.35 µg/mL) was approximately 4-fold higher than after a single dose (0.15 ± 0.06 µg/mL), but the triple dose (7.27 ± 2.32 µg/mL) was only 3-fold higher than that for the single dose.

The CPZEN-45 concentrations in BALF and lung tissue at the end of the study (5 h) are shown in Figure 6 for all treatments and dosing regimens. Five hours after administration of a single dose of powder, the concentration of CPZEN-45 in the BALF of animals dosed by INS was 0.14 µg/mL, and after passive inhalation, it was 0.33 µg/mL, indicating that at the end of the study period, there was still drug to be absorbed from the airways into the lung (Figure 6A). As expected, the concentration of CPZEN-45 in the BALF was higher in animals dosed with double and triple doses compared to those receiving a single dose, with the drug concentration in BALF increasing proportionally to the increase in the dose (Figure 6A). Moreover, at the end of the dosing period, the CPZEN-45 concentration was significantly higher in the BALF of animals dosed by passive inhalation (0.33, 0.72, and 0.84 µg/mL for the single, double, and triple doses, respectively (Figure 6A)), than that in plasma at the same time point (0.15, 0.65, and 0.45 µg/mL for the single, double, and triple doses, respectively (Figure 7)).

Among single-dosed groups, at the end of the study period, the CPZEN-45 concentration in the lungs of animals dosed by INS was significantly smaller (0.6 ± 0.38 µg/g) than that in animals dosed by passive inhalation (0.8 ± 0.65 µg/g, Figure 6B), as expected, given the higher nominal dose placed in the dosing chamber for passive inhalation compared to that by INS (80 mg per 2 animals versus 1 mg/kg per animal, respectively). Among animals receiving consecutive doses of powder by passive inhalation, at the end of the dosing period, the CPZEN-45 concentration in the lung tissue of animals receiving a double dose was more than 3-fold higher (2.93 ± 1.10 µg/g of tissue) than in the single dosed group (0.80 ± 0.65 µg/g of tissue), whereas the drug concentration in the lung tissue of animals receiving the triple dose was more than 5-fold higher (4.52 ± 2.92 µg/g of tissue) than in the single dosed group (Figure 6B). This lack of proportionality observed in plasma and tissue concentrations of animals dosed by consecutive passive inhalation doses may indicate a possible limitation in drug absorption from airway to lung tissue and from lung tissue to plasma. Most importantly, at the end of the dosing period, the CPZEN-45 concentration in the lung tissue of animals dosed by passive inhalation was significantly higher (0.80, 2.93, and 4.52 µg/g for the single, double, and triple doses, respectively (Figure 6B)) than that in plasma at the same time point (0.15, 0.65, and 0.45 µg/mL for the single, double, and triple doses, respectively (Figure 7)).

No CPZEN-45 was detected at the end of the study in spleen tissues after administration by the different routes (IV, SC, INS, or passive inhalation) or at different doses (single, double, and triple).

The PK parameters that characterize the disposition of CPZEN-45 after administration by INS and passive inhalation are presented in Table 3. The magnitude of the area under the CPZEN-45 plasma concentration versus time curve was significantly larger after administration of a “single dose by passive inhalation” (AUC_0-∞_ = 9.34 µgh/mL; *p* = 0.0095) than after administration by INS (AUC_0-∞_ = 5.76 µgh/mL) of CPZEN-45. Conversely, CPZEN-45 was eliminated at a significantly faster rate (K = 0.81 h^−1^; *p* = 0.0032) after administration of a “single dose by passive inhalation” than after INS (K = 0.39 h^−1^). Consequently, CPZEN-45 had a significantly shorter half-life (t_1/2_ = 0.86 h; *p* = 0.0037) and mean residence time (MRT = 1.39 h; *p* = 0.0065) after administration of a “single dose by passive inhalation” than after INS.

As expected, due to the high nominal dose of CPZEN-45, the maximum plasma concentration (C_max_) after administration of a “single dose by passive inhalation” was significantly higher (6.63 µg/mL; *p* = 0.0004) than that after INS, but the time to obtain this C_max_ (T_max_) was similar after both modes of administration (*p* = 0.05).

Among animals receiving consecutive doses by passive inhalation (Table 3), the magnitude of the AUC_0-∞_ was significantly larger (19.52 ± 7.97 µgh/mL) after passive inhalation of a double dose compared to a single dose (9.34 ± 1.37 µgh/mL) of CPZEN-45 powders. However, the AUC_0-∞_ after passive inhalation of a triple dose of CPZEN-45 did not increase according to the dose but was comparable (16.29 ± 5.91 µgh/mL) to the AUC_0-∞_ after passive inhalation of a double dose (19.52 ± 7.97 µgh/mL). CPZEN-45 was eliminated at the same rate after passive inhalation of a single, double, or triple dose (K = 0.81, 0.65, 0.74 h^−1^, respectively; *p* = 0.10) resulting in similar half-life (t_1/2 =_ 0.86, 1.09, 0.94 h, respectively; *p* = 0.14) and mean residence time (MRT = 1.39, 1.83, 1.78 h, respectively; *p* = 0.076). C_max_ was also statistically comparable between the three evaluated doses (6.63, 10.30, 7.27 µg/mL; *p* = 0.10), but T_max_ was significantly higher (T_max_ = 1.00 h; *p*=0.007) for double and triple doses compared to a single dose (T_max_ = 0.50 h; *p* = 0.007).

### 3.3. Disposition of CPZEN-45 after Administration of Sequential Doses by Passive Inhalation (Aerosol)

A dosing regimen consisting of three sequential “single doses of CPZEN-45 powder delivered by passive inhalation” and administered throughout a day was evaluated as an alternative to consecutive doses to avoid any potential accumulation of drug powder that could occur in the lungs of treated animals. Figure 8 shows the plasma concentrations versus time profiles after administration of the three sequential “single doses of CPZEN-45 by passive inhalation (closed circles, solid line).

To compare the disposition of CPZEN-45 between each of the sequential doses, the plasma concentration versus time profile of the first dose in the sequence was super-imposed over the second and third doses of the sequence (open triangles, dotted line). This comparison reveals that the CPZEN-45 plasma concentration for the first four time points (absorption phase and beginning of the elimination phase) are similar for the first and second doses, but the concentration in the last time point of the second dose (0.62 µg/mL) is almost 3-fold higher than that in the last time point of the first dose (0.22 µg/mL). However, after the third dose, only the plasma concentration at the first two time points matched those of the first dose. After that, all CPZEN-45 plasma concentrations were 3- to 5-fold higher after administration of the third dose, compared to those corresponding to the first dose (Figure 8), suggesting that upon administration of the third sequential dose, drug accumulation may have occurred in the lungs of the animals in this group.

CPZEN-45 concentrations in BALF and lung tissue at the end of the study period after three sequential doses (15 h) are shown in Figure 6. In contrast to the consecutive dosing regimen, the CPZEN-45 concentration in the BALF of animals receiving sequential doses (1.25 ± 0.67 µg/mL, Figure 6A) was similar to the plasma concentration at the same time point (1.50 ± 2.15 µg/mL, Figure 8). Yet, the CPZEN-45 concentration in the lung tissue of animals receiving sequential doses was significantly higher (6.50 ± 0.92 µg/mg, *p* < 0.05) than that in BALF or plasma at the same time point (Figure 6B). Notably, the concentrations of CPZEN-45 in the BALF (Figure 6A) and lung tissue (Figure 6B) of animals dosed with three sequential “single doses by passive inhalation” were significantly (*p* < 0.05) higher (1.25 ± 0.67 µg/mL and 6.50 ± 0.92 µg/mg, respectively) than that of animals receiving triple consecutive doses (0.84 ± 0.053 µg/mL and 4.52 ± 2.92 µg/mg, respectively). No CPZEN-45 was detected in the spleen tissues of animals receiving sequential doses at the end of the study period (15 h).

The disposition of CPZEN-45 after administration of three sequential “single doses of CPZEN-45 powder delivered by passive inhalation” is described numerically in terms of their PK parameters displayed in Table 4. AUC_0-∞_ appeared to be almost 4-fold larger (20.69 µgh/mL) for the third dose than for the first dose (5.67 µgh/mL). Even though clearance of CPZEN-45 was similar for all three doses (CL_ss__F = 245.07, 225.55, 256.69 mL/hkg), CPZEN-45 appeared to be eliminated faster after the first and second doses (K = 0.70 and 0.78 h^−1^, respectively) than after the third dose (K = 0.49 h^−1^). Consequently, CPZEN-45 had a shorter half-life after the first and second dose (*t*_1/2_ = 1.03 and 0.91 h, respectively) than after the third dose (*t*_1/2_ = 2.01 h).

C_max_ was also significantly higher (*p* = 0.0002) for the third dose (6.55 µg/mL) than for the first dose (3.40 µg/mL) of the sequential doses, but T_max_ was the same for all three doses in the sequential dosing regimen. The large variability in the PK parameters of individual animals, mainly observed after the third dose, made it difficult to declare statistical differences among the PK parameters of the three doses. Thus, despite the numerical differences described above, most PK parameters were not statistically different among the first, second, and third doses, except for C_max_.

## 4. Discussion

Despite the availability of effective chemotherapy, new anti-TB drugs are urgently needed to decrease the incidence of the disease that remains claiming the lives of people around the globe. New drugs should provide attractive features of having a new mode of action, short treatment time, low production cost, and activity against both drug-resistant and latent TB infection [22]. CPZEN-45 is considered to be a promising candidate because of its activity against mycobacteria [7], and since it has never been used as a therapeutic agent, bacterial resistance to this compound is not suspected.

Since it was first discovered, the overall goal has been to determine the contribution that CPZEN-45 could make to TB treatment when administered as an inhalable powder in conjunction with other oral or inhalable drugs. Progress toward this goal includes the development and optimization of a formulation of CPZEN-45 as a powder for inhalation [13,14] and preliminary studies demonstrating that pulmonary administration of this novel powder formulation as an aerosol can decrease by 1-log the bacterial load in the lungs of TB-infected guinea pigs [18]. This decrease in bacterial burden may be considered modest, and it can be attributed to several factors, including the animal model, the use of a single drug, the size of the dose, and the efficiency of the aerosol dosing procedure.

The guinea pig model of TB is considered to be a more stringent model than the mouse model of TB [23] because it develops necrotic and mineralized lesions in which viable bacteria can hide away from drug treatment [24], whereas mice mainly develop solid, non-necrotic lesions. Regardless of the animal model, treatment of infected animals with a single drug, instead of the standard drug combination, also has a limited effect on the bacterial burden of their lungs, as observed with rifampicin administered orally [25]. However, it may also be possible that the dose of CPZEN-45 powder selected from the preliminary PK study may have been insufficient to decrease further the bacterial burden in the lungs of TB-infected guinea pigs. The goal of the present study was to determine the dose and dosing regimen that can maximize the effectiveness of CPZEN-45 to treat TB.

Most recent publications describing drug distribution and permeation into different compartments where bacteria reside are performed with drugs administered orally. Extensive preclinical PK studies were performed with those drugs before being used in the clinic at doses known to have a therapeutic effect and for which the synergistic or additive effects with other drugs had already been determined. In contrast, CPZEN-45 is a new drug, and this is the first time that its PK has been described by any route and for which a therapeutic dose has not been identified. Therefore, this study can be considered as “the baseline” to determine the influence of the route of administration on the distribution of CPZEN-45, the potential therapeutic dose, and the approach to delivery. Thus, the distribution of CPZEN-45 described in the present study may be useful in defining a therapeutic dose. The administration of CPZEN-45 alone serves as a baseline for the extent to which CPZEN-45 can be expected to penetrate into the different compartments in which bacteria are located. These data can inform future drug combination studies of the addition of CPZEN-45 to maximize the therapeutic potential in treating intractable MDR and XDR TB infection.

The first step towards this goal was to determine the disposition after administration of solutions by the IV and SC routes and CPZEN-45 powder for inhalation by the pulmonary administration (INS). CPZEN-45 was absorbed faster after INS than after SC administration, as demonstrated by the initial plasma concentrations being significantly higher (Figure 2), higher C_max_, and the MAT and Ka being significantly faster (Table 2). These results agree with those obtained in guinea pigs for other drugs, including ethionamide [26], PA-824 [27], and rifampicin [28], being absorbed faster when administered by INS than orally. It is likely that the faster absorption from the lungs into systemic circulation compared to the SC or oral routes is due to the thin epithelium in the lung and the larger cardiac output into the lung [29], which enables drugs to cross the epithelium faster. In contrast, when a drug is administered orally or by SC injection, it must cross several different epithelial barriers, which delays its absorption into systemic circulation.

The advantage of pulmonary administration for CPZEN-45 is further demonstrated when comparing the drug concentrations in the plasma, BALF, and lung tissue at the end of the study period for the three different routes of administration (Figure 2 and Figure 4). At the end of the 5 h study period, the CPZEN-45 plasma concentration was higher after INS than after SC administration due to a slower rate of elimination after INS (K_e_ = 0.39 ± 0.15 h^−1^) than after SC injection (K_e_ = 0.96 ± 0.24 h^−1^), resulting in a longer *t*_1/2_ and bioavailability after INS (2.06 ± 1.01 h and 67.78 ± 31.41%, respectively) than after SC injection (0.76 ± 0.22 h and 67.78 ± 31.41%, respectively). Moreover, the CPZEN-45 concentration in the BALF and lung tissue of animals dosed by INS were significantly higher at the end of the study period compared to those after IV or SC administration (Figure 4). This suggests that even after drug levels fall below the LOD in systemic circulation, there will still be sufficient drugs to exert the antimicrobial effect in the lungs of TB-infected animals. Higher drug concentrations have also been determined at the end of the study period in the plasma, BALF, and lung tissue of animals dosed with other drugs by INS compared to systemic routes. For example, when the same dose of PA-824 was administered to guinea pigs by INS and the oral route, the drug was detected for 24 h in the plasma of animals dosed by INS, compared to only 12 h in animals dosed orally [27]. At the end of the study period (32 h), PA-824 was still detected in the BALF of animals dosed by INS, whereas no drug was detected in the BALF fluid of animals dosed orally [27]. This was consistent with a faster rate of elimination and shorter *t*_1/2_ in the orally dosed group compared to the INS group. Likewise, when the same dose of rifampicin was administered to guinea pigs by INS and the oral route, the drug was detected for 8 h in the plasma of animals dosed by INS, compared to only 3 h in animals dosed orally [30]. At the end of the study period (8 h), rifampicin was still detected in the BALF fluid and lung tissue of animals dosed by INS, whereas rifampicin was only detected in the lung tissue of animals dosed orally [30]. Yet, this was not observed for ethionamide, where the drug concentrations in plasma, BALF, and lung tissue were not significantly different between the INS and oral-dosed groups. This could be due to the differences in permeability (logP), with PA-824 and rifampicin having higher logP (2.8 and 4.0) compared to ethionamide (0.705), but given that the logP of CPZEN-45 is even lower (−0.78) than that of ethionamide, the solubility and dissolution rate of each drug in the epithelial lung fluid may also play a role, as evidenced by the fraction of the dose remaining unabsorbed in the BALF. This occurrence, when the rate of drug elimination is limited by its rate of absorption, is known as flip-flop kinetics and has been reported for compounds with limited water solubility that are delivered by the pulmonary route [28,31,32,33].

A disadvantage of pulmonary drug administration by INS is that it requires the animal to be anesthetized and intubated and the tube of the insufflator to be introduced into the trachea of the animal. For efficacy studies, in which TB-infected animals are treated daily for at least 4 weeks, direct administration of a drug by INS is impractical and can threaten the health and lives of animals. The value of INS delivery of the drug is that a known dose is delivered to which a passively inhaled drug can be referenced [34]. Ideally, aerosols of CPZEN-45 powders should be administered to conscious animals by passive inhalation, which is relevant to the method of delivery to TB-infected patients. However, drug administration by passive inhalation is a procedure with low efficiency and requires larger amounts of drugs, and the inhaled dose of aerosol is limited by the anatomical and physiological factors of the animal model that is treated. Specifically, the guinea pig is an obligate nose breather (cut-off diameter = 3 µm) with a respiratory rate of 90 breaths/min and a tidal volume of 1.8 mL/0.16 L [35]. This has an impact on the dose of aerosols that the animal is able to inhale by passive inhalation and may result in the over-estimation of the dose when extrapolating the results to humans, which can breathe both by nose and mouth and have a respiratory rate of 20 breaths/min and tidal volume 616 mL/8 L [35]. Indeed, humans would use an inhaler that requires mouth breathing to deliver the drugs on the inspiratory flow.

Another factor that limits the dose of aerosol that an animal can inhale by passive inhalation is that the concentration of drug in the aerosol is usually very diluted when commercial nose-only chambers are employed. In general, the volume of the chamber where the aerosol is being generated and from which the animals can inhale is several liters, and the aerosols are generated with the help of external air from a gas tank or a compressor, which further dilutes the drug concentration in the aerosol. Custom-made nose-only chambers have been designed to reduce the volume of the chamber and have mechanisms of aerosol generation that require little or no external air [36]. The custom-made nose-only chamber employed in this study has a volume that is 17 times smaller than that of a commercial chamber and uses the “dosators” to generate aerosols from the CPZEN-45 powders, making it very efficient and simple to achieve drug levels similar to those achieved by INS. Preliminary studies determined that eight dosators (Figure 1c), aerosolized into the chamber every 3 min, would achieve the drug levels achieved by INS [17], and this regimen was considered as a “single dose by passive inhalation” (Figure 1c). We then evaluated whether a “single dose” or consecutive “double” or “triple doses” would be able to maintain the therapeutic concentration of CPZEN-45 in relevant tissues for the 5 h study period.

CPZEN-45 appeared to be absorbed and eliminated faster when administered as a “single dose” by passive inhalation than by INS, based on the plasma concentration versus time curves (Figure 5) and the Ke (0.81 ± 0.14 h^−1^ versus 0.39 ± 0.15 h^−1^, respectively), which resulted in a shorter t_1/2_ (0.86 ± 0.15 h versus 2.06 ± 1.01 h, respectively) in plasma after passive inhalation than INS. However, these differences in absorption/elimination rates did not result in differences in the drug levels in BALF and lung tissue between INS and single-dosed groups. It is likely that the faster absorption is the result of a faster drug dissolution of CPZEN-45 in airway epithelial lining fluid (ELF) when the drug is administered over a period of 24 min instead of as a single bolus. Consequently, when the drug is absorbed faster, it can also be eliminated faster. These results with CPZEN-45 are different than those obtained with ethionamide and PA-824, where those compounds were absorbed faster and eliminated at a slower rate after INS than by passive inhalation [26,27,37]. The differences in elimination rates among these three drugs may be related to their logP, with ethionamide and PA-824 and ethionamide having a higher logP than CPZEN-45, as described above, but it may also be due to different mechanisms of elimination for each drug. The CPZEN-45 concentrations plasma, BALF, or lung tissue, as well as the AUC and C_max_, increased proportionally between a single and a double consecutive passive inhalation dose but not between single and triple consecutive doses or double and triple consecutive doses (Figure 5 and Table 3). In fact, the CPZEN-45 plasma concentration versus time curves for the double and triple consecutive doses were superimposable throughout the 5 h study period (Figure 5), whereas the concentration of CPZEN-45 in the BALF and lung tissue as well as the AUC and C_max_ for the double and triple doses were essentially the same (Table 3). This lack of proportionality between double and triple consecutive doses may be due to the saturation of the ELF of animals receiving a triple dose, where some of the undissolved and unabsorbed powder may have precipitated or cleared by alveolar macrophages [38]. These results are comparable to those obtained after INS of escalating doses of PA-824 powders [27], where there was a certain proportionality in the plasma concentration versus time curves, AUC, and C_max_ after INS of 20 and 40 mg/kg of powder. However, the plasma concentration versus time curves after INS of a 60 mg/kg dose were not significantly different than those after INS of a 40 mg/kg dose, and the C_max_ was about 20% lower than expected [27].

In contrast, the plasma concentration versus time curves and the AUC and C_max_ after the first, second, and third sequential passive inhalation doses of CPZEN-45 increased with each dose (Figure 6 and Table 4), even though each sequential dose was administered at the end of the previous 5 h study period when plasma concentrations fell below the LOQ. Specifically, C_max_ increased about 20% from the first to the second sequential dose, whereas it increased 61% from the second to the third sequential dose. Likewise, AUC increased about 28% from the first to the second sequential dose, whereas it increased 105% from the second to the third sequential dose. The increase in C_max_ and AUC after each sequential dose indicates that the drug that remained in the BALF and lung at the end of each 5 h study period (perhaps an amount similar to that after a “single dose”, blue bars, Figure 4) was still being absorbed into systemic circulation. Thus, administration of the second and third sequential doses was likely to increase the amount of drug remaining in the BALF and lungs of animals in this group (purple bars, Figure 4). These results are comparable to those obtained after sequential INS doses of capreomycin to guinea pigs, where C_max_ increased about 26% from the first to the second sequential dose, and it increased about 23% from the second to the third sequential dose. Likewise, AUC increased about 20% from the first to the second sequential dose, and it increased about 24% from the second to the third sequential dose [39]. The smaller increases in C_max_ and AUC after INS of sequential doses of capreomycin compared to passive inhalation of sequential doses of CPZEN-45 are likely due to the size of each dose and the time in between administrations. While each sequential dose of capreomycin was 20 mg/kg (approximately 10 mg per guinea pig) administered every 8 h, it is likely that the dose inhaled by each guinea pig in the custom-made nose-only chamber was much larger with a nominal dose of 80 mg aerosolized into the chamber every 5 h. In addition, the aqueous solubility of capreomycin is much higher than that of CPZEN-45, and its logP is much smaller [11,40], which could have decreased the likelihood of capreomycin being accumulated in the lungs.

Most importantly, whereas INS of CPZEN-45 maintained concentrations above the MIC in plasma for about 1.5 h and in lung tissue for about 2 h, administration of a “single dose” of CPZEN-45 by passive inhalation maintained them for about 2.5 h in plasma and administration of “double” and “triple doses” maintains them for about 3–4 h in plasma and more than 5 h in lung tissue. Notably, administration of sequential doses of CPZEN-45 in conjunction maintains plasma concentrations above its MIC in plasma for more than 10 h and in lung tissue for more than 15 h. Therefore, the best dosing regimen, from the ones evaluated in the present study, to maintain therapeutic concentrations of CPZEN-45 in the plasma and lungs of animals is the one consisting of sequential doses. Nonetheless, drug accumulation in the lungs of treated animals and its side effects must be evaluated before efficacy studies are performed with this regimen. However, a possible solution would be to reduce the size of the second and third nominal doses of powder in the regimen.

## 5. Conclusions

The results of the present study demonstrate the superiority of the pulmonary route over the SC route to administer CPZEN-45 for TB treatment. In animals, administration of CPZEN-45 by passive inhalation results in faster lung absorption than after INS. Sequential passive inhalation doses maintained therapeutic concentration levels in plasma and lung tissue for a longer time than consecutive doses; thus, for future efficacy studies, it is recommended that one nominal dose of CPZEN-45 powders be administered by passive inhalation three times per day to TB-infected animals.

## Figures and Tables

**Figure 1 pharmaceutics-15-02758-f001:**
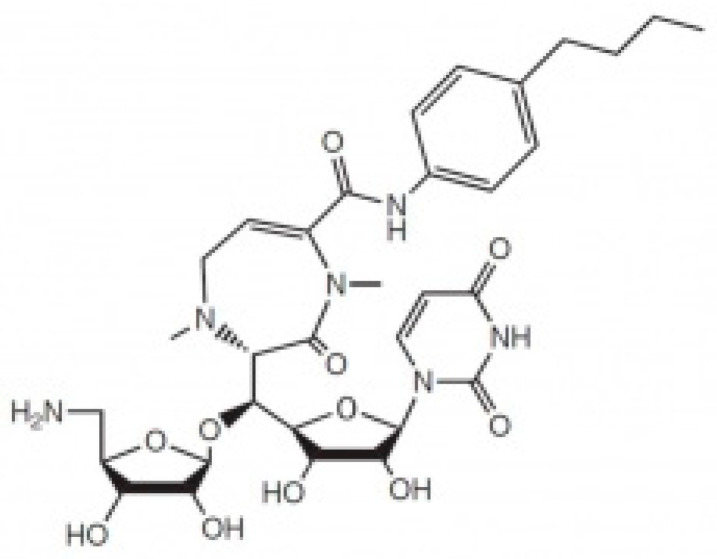
The chemical structure of CPZEN-45.

**Figure 2 pharmaceutics-15-02758-f002:**
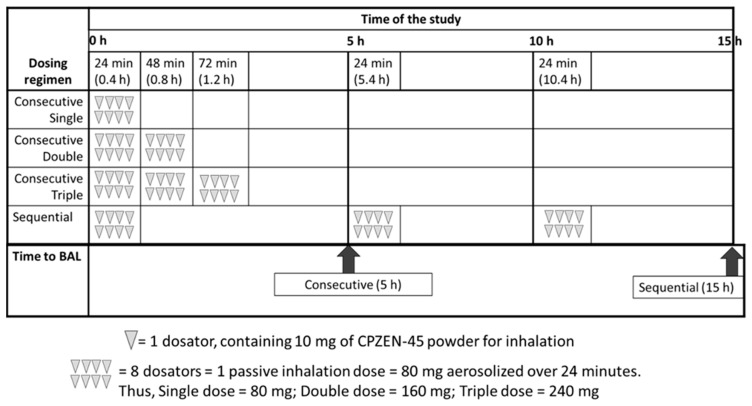
Graphical illustration of the dosing regimens employed to dose animals by passive inhalation (groups 4–7 in Table 1, as shown above).

**Figure 3 pharmaceutics-15-02758-f003:**
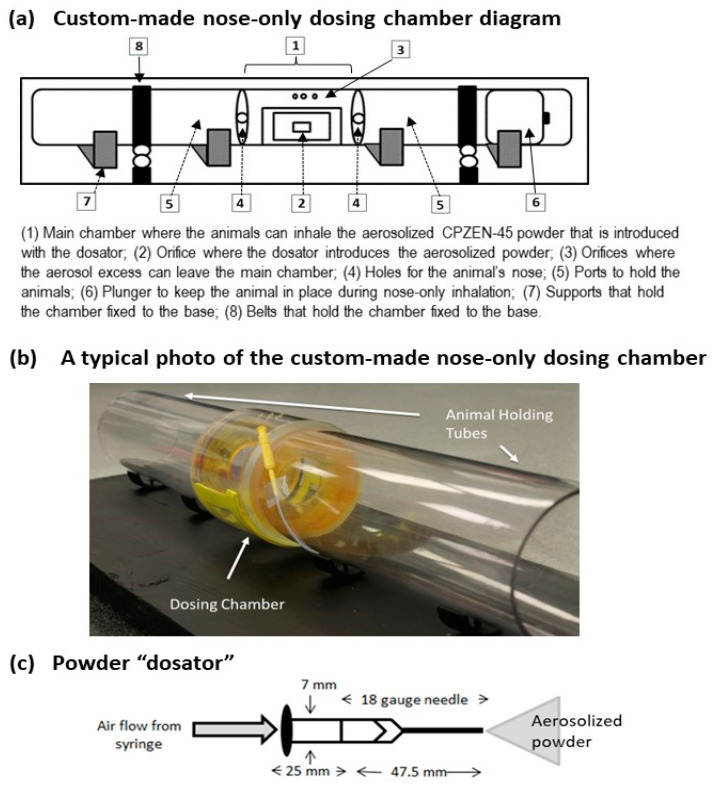
Dosing chamber (**a**,**b**) and “dosators” (**c**) employed in the present study.

**Figure 4 pharmaceutics-15-02758-f004:**
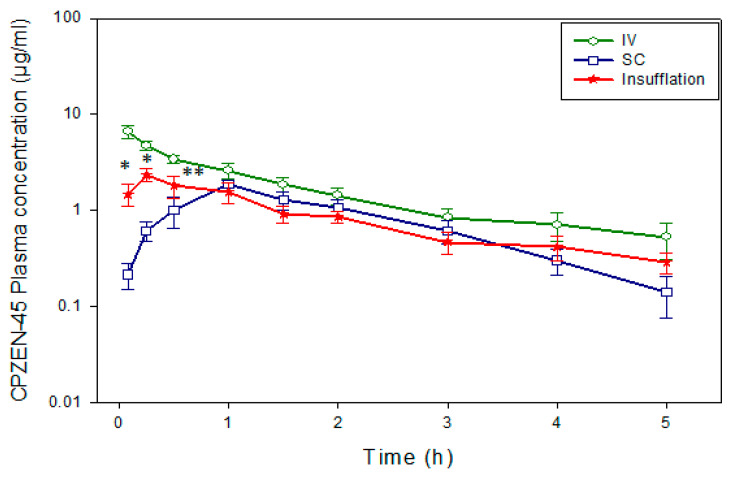
Average plasma concentration versus time curves after administration of 1 mg/kg CPZEN-45 by IV, SC, and pulmonary route (INS) routes (*n* = 6 per group). * Insufflation is significantly different than IV and SC. ** IV, SC, and insufflation are significantly different from each other.

**Figure 5 pharmaceutics-15-02758-f005:**
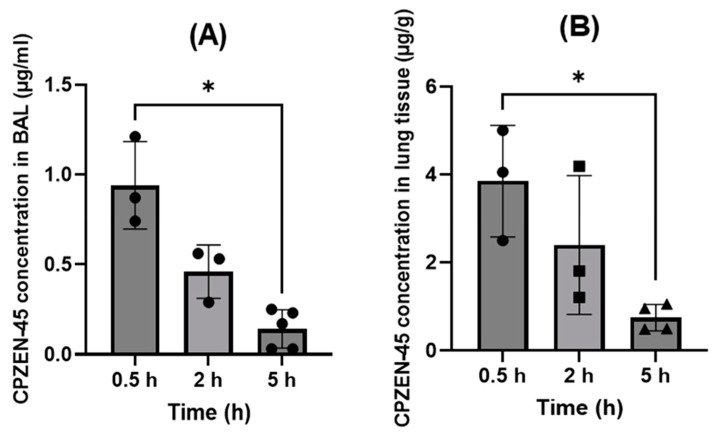
Average CPZEN-45 concentrations in (**A**) bronchoalveolar lavage (BALF) and (**B**) lung tissue at different time points after pulmonary administration (INS) of the powder (*n* = 6 per group); * Significantly different.

**Figure 6 pharmaceutics-15-02758-f006:**
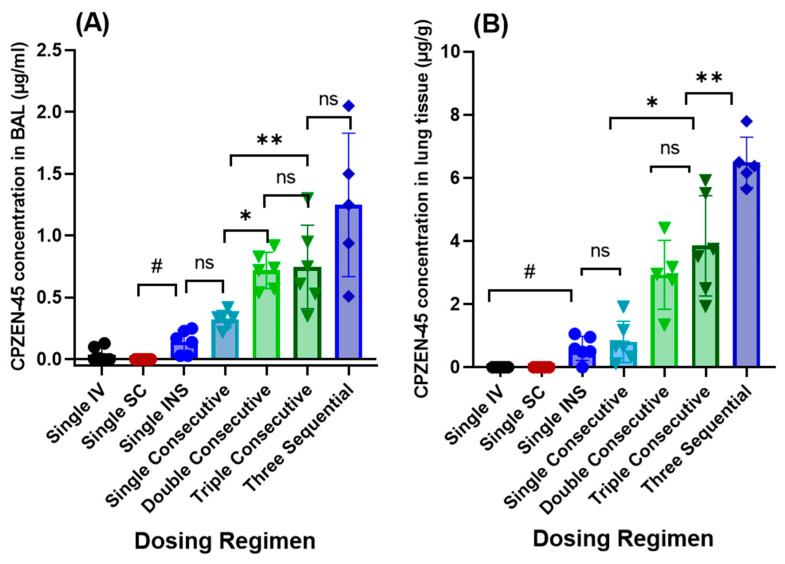
Average CPZEN-45 concentrations in (**A**) bronchoalveolar lavage (BALF) and (**B**) lung tissue at the end of the study period after administration by IV, SC, INS, and passive inhalation (*n* = 6 per group). **^#^** Insufflation achieves significantly higher lung tissue concentrations than IV or SC. * Triple consecutive doses achieve significantly higher lung tissue concentrations than single doses (*p* = 0.0087). ** Triple sequential doses achieve significantly higher lung tissue concentrations than triple consecutive doses (*p* = 0.0087). ns: no significance.

**Figure 7 pharmaceutics-15-02758-f007:**
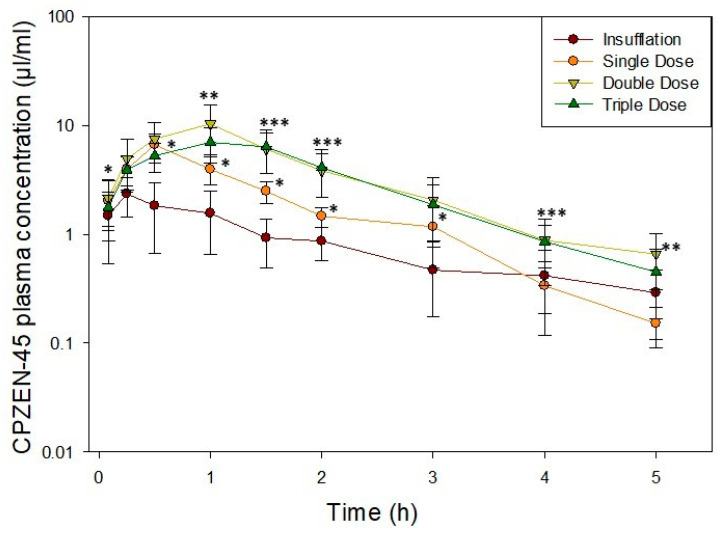
Average CPZEN-45 plasma concentration vs. time curves after administration of a single dose by INS: a single dose (8 dosators), double dose (16 dosators), and triple dose (24 dosators) by passive inhalation. * Single dose is significantly higher than insufflation. ** Double doses are significantly higher than a single dose. *** Double and triple doses are significantly higher than a single dose.

**Figure 8 pharmaceutics-15-02758-f008:**
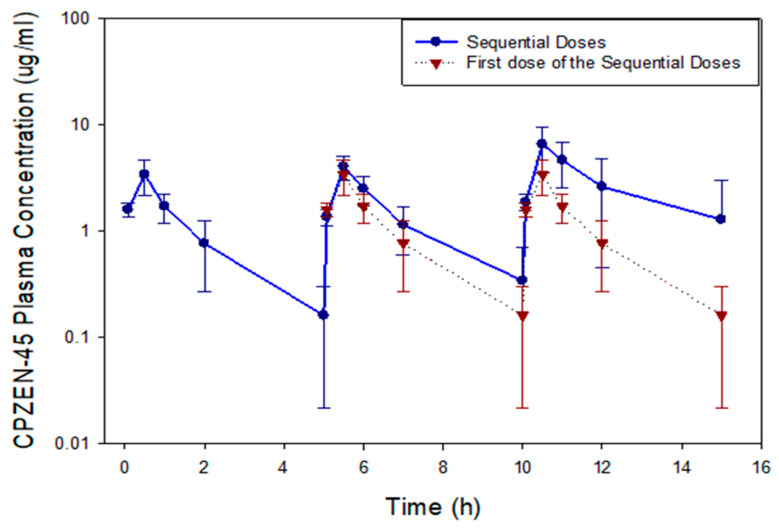
Average CPZEN-45 plasma concentration vs. time curves after administration of a single dose (8 dosators) and three multiple sequential doses (8 every 5 h) of CPZEN-45 by passive inhalation (*n* = 10).

**Table 2 pharmaceutics-15-02758-t002:** Pharmacokinetic parameters after administration of single doses of CPZEN-45 by IV, SC, and INS (mean ± SD, *n* = 6).

Parameter	Intravenous (IV) (1 mg/kg, Solution)	Subcutaneous (SC) (1 mg/kg, Solution)	Insufflation (INS) (1 mg/kg, Powder)
**Non-compartmental analysis**			
AUC (µgh/mL)	8.18 ± 2.72 ^1^	3.76 ± 1.84 ^2^	4.61 ± 1.63 ^2^
AUC_0-∞_ (µgh/mL)	8.50 ± 3.27 ^1^	4.05 ± 2.01 ^2^	5.76 ± 2.64 ^1,2^
CL_SS__F (ml/hkg)	136.07 ± 43.30 ^2^	277.16 ± 121.09 ^1^	236.70 ± 74.75 ^1,2^
K_e_ (h^−1^)	0.64 ± 0.20 ^2^	0.96 ± 0.24 ^1^	0.39 ± 0.15 ^2^
*t*_1/2_ (h)	1.14 ± 0.27 ^2^	0.76 ± 0.22 ^2^	2.06 ± 1.01 ^1^
MRT (h)	1.66 ± 0.39 ^2^	1.90 ± 0.37 ^2^	2.83 ± 0.84 ^1^
MAT (h)	-	0.42 ± 0.17	1.17 ± 0.84
C_max_ (µg/mL)	6.89 ± 2.06 ^1^	1.91 ± 0.48 ^2^	2.35 ± 1.02 ^2^
Tmax (h)	0.08 ± 0.00 ^3^	0.83 ± 0.25 ^1^	0.37 ± 0.18 ^2^
F__AUC_	-	46.01 ± 22.49	56.39 ± 19.94
F__AUC0-∞_	-	47.73 ± 23.72	67.78 ± 31.14
**One Compartment analysis**			
K_a_ (h^−1^)	-	1.23 ± 0.55	12.94 ± 5.66
K_e_ (h^−1^)	0.43 ± 0.19 ^2^	0.95 ± 0.35 ^1^	0.47 ± 0.13 ^2^

Numeric superscripts show the relative ranks of values. When the means are not statistically different, the same numerical superscript is used.

**Table 3 pharmaceutics-15-02758-t003:** Pharmacokinetic parameters after administration of single doses of CPZEN-45 by INS; single, double, and triple doses by passive inhalation (mean ± SD, *n* = 5–6).

Parameter	Insufflation (INS) (1 mg/kg)	Aerosol Single Dose (80 mg)	Aerosol Double Dose (160 mg)	Aerosol Triple Dose (240 mg)
AUC (µgh/mL)	4.61 ± 1.63 ^2^	9.14 ± 1.43 ^2^	18.41 ± 7.72 ^1^	15.65 ± 5.48 ^1,2^
AUC_0-∞_ (µgh/mL)	5.76 ± 2.64 ^1,2^	9.34 ± 1.37 ^2^	19.52 ± 7.97 ^1^	16.29 ± 5.91 ^1,2^
CL_SS__F (ml/hkg)	236.70 ± 74.75 ^1,2^	215.68 ± 27.42 ^1^	247.17 ± 134.43 ^1^	226.20 ± 56.19 ^1^
K_e_ (h^−1^)	0.39 ± 0.15 ^2^	0.81 ± 0.14 ^1^	0.65 ± 0.12 ^1^	0.74 ± 0.12 ^1^
*t*_1/2_ (h)	2.06 ± 1.01 ^1^	0.86 ± 0.15 ^2^	1.09 ± 0.26 ^2^	0.94 ± 0.12 ^2^
MRT (h)	2.83 ± 0.84 ^1^	1.39 ± 0.27 ^2^	1.83 ± 0.37 ^2^	1.78 ± 0.26 ^2^
C_max_ (µg/mL)	2.35 ± 1.02 ^2^	6.63 ± 1.80 ^1^	10.30 ± 4.90 ^1^	7.27 ± 2.32 ^1^
Tmax (h)	0.37 ± 0.18 ^2^	0.50 ± 0.00 ^2^	1.00 ± 0.00 ^1^	1.00 ± 0.44 ^1^

Numeric superscripts show the relative ranks of values. When the means are not statistically different, the same numerical superscript is used.

**Table 4 pharmaceutics-15-02758-t004:** Pharmacokinetic parameters after passive inhalation of single and three sequential doses of CPZEN-45 (mean ± SD, *n* = 4–6).

Parameter	Aerosol 8 × 8 × 8 Doses (5 h, 80 mg)	Aerosol 8 × 8 × 8 Doses (10 h, 80 mg)	Aerosol 8 × 8 × 8 Doses (15 h, 80 mg)
AUC (µgh/mL)	5.40 ± 2.65 ^1^	6.89 ± 2.27 ^1^	14.14 ± 9.82 ^1^
AUC_0-∞_ (µgh/mL)	5.67 ± 2.89 ^1^	7.29 ± 2.45 ^1^	20.69 ± 21.00 ^1^
CL_SS__F (mL/hkg)	245.07 ± 100.80 ^1^	225.55 ± 77.26 ^1^	256.69 ± 156.67 ^1^
K_e_ (h^−1^)	0.70 ± 0.22 ^1^	0.78 ± 0.20 ^1^	0.49 ± 0.32 ^1^
*t*_1/2_ (h)	1.03 ± 0.25 ^1^	0.91 ± 0.24 ^1^	2.01 ± 1.51 ^1^
MRT (h)	1.39 ± 0.22 ^1^	1.65 ± 0.38 ^1^	2.92 ± 2.19 ^1^
C_max_ (µg/mL)	3.40 ± 1.25 ^2^	4.05 ± 1.02 ^1,2^	6.55 ± 3.05 ^1^
Tmax (h)	0.50 ± 0.00 ^1^	0.50 ± 0.00 ^1^	0.50 ± 0.00 ^1^

Numeric superscripts show the relative ranks of values. When the means are not statistically different, the same numerical superscript is used.

## Data Availability

This work was presented in the abstract as a poster and selected oral presentation “Pharmacokinetics of CPZEN-45, a novel anti-tuberculosis drug in male guinea pigs.” at the International Society for Aerosols in Medicine, Biannual Meeting in Chapel Hill, NC, USA, 6–10 April 2013.

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
