# Peer review of "The Pharmacokinetics of CPZEN-45, a Novel Anti-Tuberculosis Drug, in Guinea Pigs"

_pharmaceutics, 2023, doi:10.3390/pharmaceutics15122758_

Round 1

Reviewer 1 Report

Comments and Suggestions for Authors

The authors have made significant advancements in elucidating the pharmacokinetic attributes and dosing regimen pertinent to CPZEN-45, an innovative inhalable compound intended for the treatment of tuberculosis in guinea pigs. Their endeavors have yielded notable insights into the manner by which CPZEN-45 is absorbed, distributed, and made bioavailable through diverse routes of administration, encompassing intravenous, subcutaneous, and direct pulmonary modalities. Additionally, they have probed into the fate of CPZEN-45 powders subsequent to passive inhalation, employing both consecutive and sequential dosing methodologies. While the study bears a comprehensive nature, the inclusion of supplementary experiments may amplify its depth and influence:

Conducting histological analyses stands to illuminate the intricate interplay between CPZEN-45 and tuberculosis-affected lung tissues, thereby unveiling potential loci of action and any distinctive tissue-specific accumulation patterns. Extending the scrutiny of CPZEN-45's distribution beyond pulmonary confines promises a holistic grasp of its pharmacokinetic dynamics. Quantifying concentrations in organs such as the liver, spleen, and lymph nodes holds the promise of shedding light on prospective off-target effects.

In its entirety, the manuscript boasts a well-structured disposition and an adeptly articulated narrative, thus undoubtedly meriting its consideration for publication.

Author Response

The authors have made significant advancements in elucidating the pharmacokinetic attributes and dosing regimen pertinent to CPZEN-45, an innovative inhalable compound intended for the treatment of tuberculosis in guinea pigs. Their endeavors have yielded notable insights into the manner by which CPZEN-45 is absorbed, distributed, and made bioavailable through diverse routes of administration, encompassing intravenous, subcutaneous, and direct pulmonary modalities. Additionally, they have probed into the fate of CPZEN-45 powders subsequent to passive inhalation, employing both consecutive and sequential dosing methodologies. While the study bears a comprehensive nature, the inclusion of supplementary experiments may amplify its depth and influence:

Conducting histological analyses stands to illuminate the intricate interplay between CPZEN-45 and tuberculosis-affected lung tissues, thereby unveiling potential loci of action and any distinctive tissue-specific accumulation patterns.

The histological analysis of the lungs of animals treated in efficacy studies has been published previously (Reference #16 in the present manuscript). The lungs of animals treated with aerosolized CPZEN-45 appeared normal in regions that were not granulomatous, and there was no evidence of drug accumulation.

Extending the scrutiny of CPZEN-45's distribution beyond pulmonary confines promises a holistic grasp of its pharmacokinetic dynamics. Quantifying concentrations in organs such as the liver, spleen, and lymph nodes holds the promise of shedding light on prospective off-target effects.

In addition to plasma, BAL and lung tissue, the concentrations of CPZEN-45 were determined in the spleen of treated animals and no drug was detected in the spleen at the time points evaluated (0.5, 2 and 5 hours). A more comprehensive determination of the presence of CPZEN-45 in other tissues will be evaluated in formal toxicological studies.  

In its entirety, the manuscript boasts a well-structured disposition and an adeptly articulated narrative, thus undoubtedly meriting its consideration for publication.

We appreciate the kind words of the reviewer.

Reviewer 2 Report

Comments and Suggestions for Authors

The manuscript investigates the pharmacokinetics of a novel anti-tuberculosis drug in guinea pigs, with a focus on the comparative analysis of different administration routes, namely intravenous (IV), subcutaneous (SC), and direct pulmonary administration(INS) . The study contributes valuable insights into the drug's delivery within the guinea pig model, shedding light on potential avenues for optimizing treatment regimens and clinical translation. I would like to commend the authors on a well written manuscript and have only a small number of comments or suggestions to improve the overall quality of the manuscript.

I would recommend including a small section in the introduction to discuss the existing rationale for choosing one route of administration over the other other options and why direct pulmonary administration may be favoured over the other routes based on other literary evidence for this particular treatment which can then be supported by the results.

The introduction could also look at the other emerging compounds being investigated TB and the similarity / difference to CPZEN-45

Figure 1a - A legend or labels should be provided to identify the relevant parts of the apparatus.

Figures 2, 5 and 6 - Resolution of each graph needs to be improved.

Figure 3 - Can the authors provide an explanation for the number of data points in each graph as they are not consistent with the claimed n=6 per group.

All equations should be numbered sequentially.

Can you please check with the editor whether the the "t" in "table" and the "f" in "figure" are capitalised when referring to a specific table or figure created in the main body of the manuscript. (Some journals do not follow this convention, but most in the MDPI family do.)

Author Response

The manuscript investigates the pharmacokinetics of a novel anti-tuberculosis drug in guinea pigs, with a focus on the comparative analysis of different administration routes, namely intravenous (IV), subcutaneous (SC), and direct pulmonary administration (INS). The study contributes valuable insights into the drug's delivery within the guinea pig model, shedding light on potential avenues for optimizing treatment regimens and clinical translation. I would like to commend the authors on a well written manuscript and have only a small number of comments or suggestions to improve the overall quality of the manuscript.

I would recommend including a small section in the introduction to discuss the existing rationale for choosing one route of administration over the other options and why direct pulmonary administration may be favoured over the other routes based on other literary evidence for this particular treatment which can then be supported by the results.

The introduction could also look at the other emerging compounds being investigated TB and the similarity / difference to CPZEN-45

We appreciate the suggestion. These elements have now been incorporated in the introduction.

Figure 1a - A legend or labels should be provided to identify the relevant parts of the apparatus.

These have now been added to the figure.

Figures 2, 5 and 6 - Resolution of each graph needs to be improved.

We agree that this makes the image more clear, and thank the reviewer for the suggestion. The figure has been modified as suggested.

Figure 3 - Can the authors provide an explanation for the number of data points in each graph as they are not consistent with the claimed n=6 per group.

We thank the reviewer for identifying this inconsistency which resulted from an error in transcription. These numbers were carried from figure 2 as we copy-pasted the legend from figure 2 into figures 3 and 4. The correct number animals per group are now stated in the individual figures.

All equations should be numbered sequentially.

The equations have been numbered in the revised manuscript as directed.

Can you please check with the editor whether the the "t" in "table" and the "f" in "figure" are capitalised when referring to a specific table or figure created in the main body of the manuscript. (Some journals do not follow this convention, but most in the MDPI family do.)

We are grateful for the suggestion. We looked at recently published articles and they do capitalize these words in the text. Thus, we have now capitalized these words in our manuscript.

Reviewer 3 Report

Comments and Suggestions for Authors

This article reported a pharmacokinetics study about a novel compound. As the compound was newly discovered, and the indication was for anti-tuberculosis, the topic was to some degree of significance. The paper fell within the scope of Pharmaceutics, and might arouse a certain impact in its field. However, there were still some flaws. Therefore, the reviewer suggested to accept this paper after a Major Revision. Please refer to the detailed comments:

1.       If not restricted by a patent, it was strongly recommended to provide the structure or synthetic method of the compound CPZEN-45.

2.       In addition to the synthetic methodology, the preparation method of CPZEN-45 powder should be described. Please notice that this was what the audience of Pharmaceutics most concerned about.

3.       The ethical approval number for the animal studies should be listed.

4.       Could the authors offer a typical photo of the dosing chamber?

5.       The formulae should be encoded as Formula 1, Formula 2, etc.

6.       The last time point of the PK study was 5 h. The whole experimental period was quite short. Why not try a 12-h or 24-h period?

7.       In some figures like Figure 3 and 4, the authors denoted that n = 6. However, there were only 3~5 scatters in some groups. Please carefully explain the reason.

8.       It was found that the authors used log y axis throughout the text. Compared to normal y axis, log y axis might sometimes “magnify” the difference between small values and “reduce” the difference between large values. Please, if possible, show the results with normal y axis.

Round 2

Reviewer 1 Report

Comments and Suggestions for Authors

The authors have addressed my concerns, and the manuscript can be accepted for publication. Good luck!

Reviewer 3 Report

Comments and Suggestions for Authors

Thanks for your revision.